# Design and Metrological Analysis of a Backlit Vision System for Surface Roughness Measurements of Turned Parts

**DOI:** 10.3390/s23031584

**Published:** 2023-02-01

**Authors:** Alessia Baleani, Nicola Paone, Jona Gladines, Steve Vanlanduit

**Affiliations:** 1Department of Industrial Engineering and Mathematical Sciences, Università Politecnica delle Marche, 60121 Ancona, Italy; 2Faculty of Applied Engineering, Universiteit Antwerpen, 2000 Antwerp, Belgium

**Keywords:** surface roughness measurement, backlit vision-based measurement system, non-contact measurement system, uncertainty analysis

## Abstract

The focus of this study is to design a backlit vision instrument capable of measuring surface roughness and to discuss its metrological performance compared to traditional measurement instruments. The instrument is a non-contact high-magnification imaging system characterized by short inspection time which opens the perspective of in-line implementation. We combined the use of the modulation transfer function to evaluate the imaging conditions of an electrically tunable lens to obtain an optimally focused image. We prepared a set of turned steel samples with different roughness in the range *R_a_* 2.4 µm to 15.1 µm. The layout of the instrument is presented, including a discussion on how optimal imaging conditions were obtained. The paper describes the comparison performed on measurements collected with the vision system designed in this work and state-of-the-art instruments. A comparison of the results of the backlit system depends on the values of surface roughness considered; while at larger values of roughness the offset increases, the results are compatible with the ones of the stylus at lower values of roughness. In fact, the error bands are superimposed by at least 58% based on the cases analyzed.

## 1. Introduction

Sustainable and effective production is the drive of the somewhat new and ever-evolving trend of zero defect manufacturing (ZDM). ZDM aims at eliminating production scraps, not only through detecting and correcting defective products but also through defect prediction and prevention [1].

The idea of a ZDM production was introduced in the 1960s, but it was only a few years later, with the arrival of Industry 4.0 that the development of new enabling technologies, such as in-line data gathering and digital twins, allowed ZDM to move forward [2].

Today, this means that quality control of the finished product has moved from the end of the production line to being distributed along the production line itself. This is mandatory in order to develop a strategy that involves the prediction and correction of defects by exploiting early detection of defects and trends, as well as through data sharing between processes which allows for dealing with streams of variations [3]. A shift in quality approach is mandatory, since the new strategies need data to be collected directly from the process and the product in order to obtain a close integration of process and quality control and to achieve in-line testing on 100% of production [4,5]. The challenge in many cases is to have measurement systems applicable for in-line testing on 100% production that operate fast enough to cope with the production rate and robust enough for a reliable operation in a harsh environment. For this scope, non-contact measurement systems are preferable; imaging technologies offer interesting perspectives for developing suitable solutions. Given this, in many industrial realities, measurements for quality monitoring are still based on a statistical approach, where a random sample is picked from the production line and checked for defects in a quality control laboratory. This is particularly true in batch production, where no individual part tracking is implemented.

An important aspect to consider is the uncertainty associated with measurements, which can be caused, among other reasons, by the noisy environment of the shop floor or by the influence of human operators. In fact, measurements performed by workers are subject to a relatively large uncertainty.

It is obvious that the quality of the data depends entirely on the uncertainty of measurement, which in turn affects the reliability of the diagnosis made on measured data. This is why in conformity assessment one of the main focuses is to reduce uncertainty to the minimum level with respect to the acceptable tolerance interval. For example, in this perspective, ISO standards such as ISO 14253-1:2017 prescribe measurement uncertainty to be a small fraction of the tolerance interval whereby small is intended to be at least one-fifth [6].

In the manufacturing industry that produces metal parts, turning is one of the main production processes and it requires a high velocity and precision, especially for high-precision mechanical parts. When producing parts in batches, usually the operators are in charge of performing dimensional measurements in metrological laboratories. Dimensions and surface quality are the characteristics they check to determine compliance to specifications of the finished product, surface quality is assessed by measuring the roughness of the surface [7]. However, surface roughness highly depends on the production process and on the tool wear. A larger surface roughness can negatively influence the properties of the workpieces, such as appearance, reliability and function [8], and can be caused by a damaged or worn tool. To counter this effect, the tool is replaced regularly, and the surface roughness is periodically checked.

In line with the trend towards ZDM, automation of dimensional metrology and surface roughness metrology has become a main part of manufacturing and quality control processes [9], since in-line quality control is one of the pillars of digital factories and Industry 4.0.

State-of-art measurements of surface roughness are carried out using contact measurement systems and can be implemented only off the production line. Therefore, such measurements are used in the context of statistical process control and batch production. Indeed, surface roughness measurements can be divided into two main categories: contact techniques and non-contact techniques. The first usually involves use of a stylus with a diamond tip which is scanned along a straight line over the surface and acquires the deviation in the form of a one-dimensional surface profile [10].

This method, even though it is still commonly used, has some limitations:Workers have to randomly select a sample from the production line to test it with the stylus;The stylus inspects only a relatively small area, which does not represent the whole surface;The sample could be scratched by the stylus such that its local mechanical properties are compromised. In fact, the stylus tip is pressed against the surface with such a force that allows it to remain in contact with the surface under measurement during the transducer movement [11];The stylus tip radius represents a resolution limit of the instrument, since the finite size of the stylus tip results in some loss of information [12];Overall, it is a method usable only off-line and not suitable to test 100% of production for large lot sizes.

Non-contact measurements are a topic of interest and different techniques have been explored to perform roughness measurements, among which there are:Shearing interferometry: a microscopy technique used to obtain images of samples with small height deviations, it consists of overlapping two images of an object which are shifted laterally relatively to each other [13];Atomic force microscopy (AFM): is based on the atomic scale repulsive or attractive forces on a sprung cantilever. A diode laser is focused on the cantilever tip and as the tip scans the surface the laser beam is deflected onto a photodiode. Hence, the light beam also changes position. Its resolution is limited to tens of micrometers and a roughness measurement with AFM requires multiple scans at different locations of a sample surface. To apply this technique, sample cleanliness must be ensured to avoid artefacts and AFM is sensitive to the surrounding environment [14,15];Scatterometry: the phase changes of light reflected from the surface are detected and used to extract information about the shape of the surface. Measuring surface roughness through scatterometry requires careful modeling and re-constructing of signals. Moreover, samples must be perfectly flat and with roughness lower than 5 nm [15,16]. The literature also reports some attempts to apply scatterometry for in-line measurements [17,18];Laser speckle photography: the contrast of the speckle image is used to trace back to surface roughness. It is needed a translation of the surface or the detector to have an intensity of the speckle field high enough for the calculation. This real time technique has a range of measurement limited to less than 1 µm [19];Digital holography: based on optical interference and diffraction. It uses holograms to obtain the intensity and the phase of an object [20].

Even though on one hand optical methods are a more complex technology, on the other hand they have some inherent advantages [21]:The information content is high because processing an image allows obtaining spatial information without scanning;Measurements are fast and non-contact;Their non-contact nature leaves no scratches on the sample.

However, optical methods are more sensitive to variables such as optical constants and surface features [13]. Moreover, almost none of these methods can be implemented for measurements in-line during production since they are methods that are designed for and work best in laboratory conditions.

Machine vision methods and vision-based methods in general are less sensible to disturbances and are apt to measure surface roughness. In fact, they have attracted some research interest, as reported in the literature [21,22,23,24,25,26,27,28,29].

In this work we presented the design and performed a feasibility study of a high-magnification backlit vision-based measurement system [30] conceived to perform dimensional measurements at the microscale, in particular to measure surface roughness during in-line production. The system is designed to take images during turning and process them to determine surface roughness. Since image acquisition is a very fast and non-contact process, the system would enable the possibility of shortening inspection time and completely non-contact measurement, which would appropriately fit the requirements of ZDM, i.e., in-line quality control of 100% of production on moving/rotating parts.

A scheme of how it works is presented in Figure 1: a turned sample (a cylinder) is placed in front of a camera, its axis is orthogonal to the camera axis, the camera axis is aligned tangent to the cylinder, and it is backlit by a collimated source of light. This optical configuration is typical of telecentric vision [31] or backlit vision [32] and it provides the image of the edge of the sample.

If magnification is sufficiently large, the acquired image allows resolution of surface roughness which appears as a black wavy profile over a white background. During image processing, the image is analyzed, the profile is extracted and the average surface roughness is measured as the average of the distances of the peaks and valleys from the mean line.

In order to carry out development of the system we divided the research into two steps: first is design and analysis of performance in lab conditions. Once the performance is assessed and optimized, the system will be integrated in a production line lathe machine for validation in operating conditions.

This paper presents the first phase of the research.

In Section 2 the design of the instrument is presented and the measurement algorithm is described. Optimization of the system parameters is reported and it is explained how the measurement campaign was planned, how the test samples were prepared and which instruments were used to perform a comparison. Section 3 reports the results of the study. Results are discussed in Section 4, with attention focusing on critical problems we found related to the location of measurement and the resolution of the stylus. Lastly, we conclude with Section 5.

## 2. Materials and Methods

### 2.1. Measurement System Setup

The instrument was designed by selecting the components of the imaging part and of the illumination part in order to achieve the desired performances in terms of field of view (FoV) and spatial resolution suitable to measure surface roughness in a range of approximately 2–15 µm. The addition of a tunable lens allows electronic control of the focus. All components were selected from commercially available parts taking into account cost–performance ratio in order to be suitable for future industrialization.

The measurement system setup consists of two main parts, the imaging setup and the back-lighting system (see Figure 2A). The imaging part is made of:An objective composed of two parts:
An electrically tunable lens (ETL) EL-10-30 C (Optotune);A dynamic focus VZM lens (Edmund Optics) with a magnification range of 0.65×–4.6×;A 5 Mpix camera with a “2/3” sensor (Lucid Vision Labs Triton, TRI051S-MC).

The ETL was useful to optimize the focus without having to readjust the reciprocal position of the sample and the vision system each time. The ETL is made of two elastic polymer membranes which are filled with an optical fluid, and the deflection of the lens is proportional to the pressure in the fluid. To exert pressure on the fluid, there is an electromagnetic actuator, hence by applying current to the actuator it is possible to change the focal distance of the lens in a matter of milliseconds.

The wide magnification range of this design grants a very small FoV. In our configuration we obtained an approximate FoV of 1.5 mm by 1.8 mm using 4.6× magnification power. The variation in focus given by the ETL also created a slightly variable FoV of the system, ranging from 1.4 mm by 1.6 mm to 1.5 mm by 1.8 mm. This FoV is imaged onto the 5 Mpix “2/3” sensor, thus allowing for the necessary spatial resolution; pixel dimensions projected over the object plane resulting between 0.6 µm/pixel and 0.7 µm/pixel at the maximum magnification of 4.6×. The depth of field of the imaging objective results are 84–90 mm at the maximum magnification of 4.6×.

As for the backlighting of the vision measurement system, it is made up of two parts:A white 3W LED;A lens with focal length of 150 mm.

This created the parallel collimated illumination needed for this application to have a sharp edge instead of the blurred shadow that would arise if uncollimated light were used. The illumination intensity is uniform across the image, resulting in a flat-top profile. In fact, over an 8-bit image intensity scale (0–255), the average illumination is in the range 150–200 with a dispersion lower than three estimated as standard deviation.

In Figure 2A a schematic of the instrument is represented, while Figure 2B is an image of the actual measurement setup.

Even if industrialization of this idea is out of the scope of this paper, such a conceptual scheme allows the realization of an optical instrument that could be designed for an in-line application in a lathe. The parallel light projector should be placed on one side of the turning piece, while the imaging objective and the camera should be on the other side. To remove image blur induced by motion, light should be pulsed and camera acquisition shuttered at proper aperture time. This would be useful if an in-process roughness measurement is desired. As a second option, the system could be placed by the lathe, and turned parts could be picked up by a robotic arm and placed in the measurement system. This would allow a measurement by-the-process. In both cases, this would be fully in line with the scope of having measurements on 100% of parts produced, as required by ZDM.

### 2.2. Measurement Algorithm

The target images are a sharp close-up view of the edge of the sample, where the wavy shadows created by the machining process are clearly visible, see Figure 3. In the figure are depicted two different samples with different surface roughness. The image on the left is of a sample with low surface roughness (*R_a_* = 2.4 µm), while the one on the right has much higher surface roughness (*R_a_* = 15.1 µm). At the edge of the image, a light scattering effect is visible. This will be discussed in Section 2.3.3.

The measurement algorithm is based on four steps (see also Figure 4):An edge-preserving filter (i.e., median filter) is used to remove noise from the 8-bit grey-level image;The image is scanned from grey to black with a Sobel algorithm to detect the edge of the sample [33]. It detects edges in both directions and it finds edges where gradient is maximum. To this end, the edge function in Matlab was used;To determine the mean line and to detrend the mean surface, the edge of the piece is linearly fitted by using a least square fit model of the type *y = ax + b*;The average surface roughness, *R_a_*, is calculated as the arithmetic mean of the absolute ordinate values within a sampling length [34]:
(1)$$R_{a}=\frac{1}{n}\overset{n}{\underset{i=1}{\sum }}\left|Z_{i}\right|$$
where *n* is the number of data points (i.e., the sampling length) and *Z_i_* is the ordinate value of the *i*th point from the mean line (see Figure 4, step 4).

### 2.3. Optimization of System Parameters

#### 2.3.1. Calibration

In order to calibrate the vision system, a miniature reference calibration target with 50 µm by 50 µm squares was used.

The variability of the FoV is an effect of the combination of all the optics. Its change depends on the ETL used. In fact, the conversion factor between mm and pixel would change every time the current applied to the ETL changed. To solve this problem, a calibration function correlating ETL current and conversion factor was experimentally built, based on the knowledge that the relationship between the two is linear. This would also indirectly tie the ETL current and the FoV.

Figure 5 shows the results of the calibration and proves the linear correlation between ETL current and FoV.

Six images of a calibration target were collected at different but known current values of the ETL, placing the target in focus by moving it manually and not changing the current. The conversion factor was extracted for each image by correlating the number of pixels to the known dimensions of the squares. This way we obtained six pairs of values on which we reconstructed a linear function. Once we had a function correlating the conversion factor to the input current of the ETL, for each measurement we recalculated the conversion factor based on the ETL current of that specific case.

The current interval needed to place into focus the samples during measurements was between 78.4 mA and 165.6 mA, hence the relative computed range of conversion factors resulted to be between 0.6 µm/pixel and 0.7 µm/pixel in both the x and y directions of the camera sensor, with a standard deviation of 5.7 × 10^−5^ µm/pixel.

#### 2.3.2. Modulation Transfer Function

The modulation transfer function (MTF) is a parameter which represents the sharpness of an imaging system, making it possible to evaluate the imaging conditions and to assess the performance of the optical system. It is also known as the spatial frequency response of an imaging system to an input, therefore is directly related to spatial resolution of the imaging system. The amplitude of MTF can vary between 100% and 0%, with one indicating a perfect preservation of contrast [35]. The reference parameter is represented by the MTF at 50%, which indicates the highest line frequency that can be replicated by a lens without allowing the MTF to go lower than 50%. In fact, this is the value related to perceived image sharpness.

Among the possible methods to evaluate the MTF we chose the slanted edge method [36] where a step input is given to the imaging system through the image of a sharp slanted edge (see Figure 6 and Figure 7A), the system’s response is then the edge spread function (ESF) which is then derived to obtain the line spread function (LSF). Lastly the MTF is the Fourier transform of the LSF [37]. To perform this analysis we used the Slanted Edge MTF Plugin of ImageJ open-source software package [37] because it is easily available since it is open-source and its known limitations (it does not perform well on noisy images) do not apply to our case.

Based on the MTF, the right combination of exposure time and intensity of illumination was determined in order to obtain the highest contrast that would not cause saturation. This would cause blooming, hence a change in charge distribution on the camera sensor. This affects profile shape and position which should be avoided when performing geometric measurements. The exposure time thus selected was the one that would increase the band width the most and the obtained preservation of contrast was of 0.058 cycles/pixels.

#### 2.3.3. Threshold

If we look at the edge of the sample, it is evident that there is light scattering caused by the collimated impinging light interacting with the surface that in turn produces diffraction fringes. The phenomenon appears as fringes parallel to the edge. In diffraction, fringe spacing depends on wavelength λ; therefore, fringes will be sharp only with monochromatic illumination. In Figure 8, it is presented an intensity profile orthogonal to the surface (Figure 8A); we see a transition from black to the first peak of the diffraction fringe followed by other smaller oscillations (Figure 8B). If we had used a monochromatic light source the diffraction fringes would be clearly distinct and sharp, while if we had used a wide-spectrum light source they would appear as a fuzzy blur due to the superposition of fringes with different geometry. In our case, the LED source is neither monochromatic nor a continuous spectrum; its spectrum has a peak in the 450–460 nm range, but it also covers the range of 480–700 nm, hence we are in between the blurry fringes and the distinct fringes.

Overall, the edge is not sharp. Nonetheless, the threshold-based edge detection algorithm is able to accurately detect the edge of the sample. The value associated with the threshold is the cut-off value between black and white pixels in a binary representation of the image intensity. In Figure 9 the effect of different thresholds is represented. It can be seen that the shape of the profile is maintained and only shifted either towards the black side or towards the grey side. In fact, even if the transition from black to grey is not sharp, a difference in threshold would only cause a shift in the detected profile as schematized in Figure 8B: the intersection between the threshold and the transition profile from black to grey is only offset towards the right.

The position between the mean lines changes only by 1 pixel. This difference is also coherent with the offset of the profile lines which also differ by 1–2 pixels. If a change in threshold causes a shift in the profile, the estimate of *R_a_* does not vary, since *R_a_* is a quantity that describes the shape of the profile, not its actual position. The *R_a_* values corresponding to the three thresholds (S1, S2, S3) are, respectively: *R_a_* = 16.1 µm, *R_a_* = 16.6 µm and *R_a_* = 16.9 µm.

Another issue which usually affects the performance of optical instruments is measurement noise depending on random fluctuations of illumination and image sensor response [38,39]. This noise typically affects a measurement method based on intensity. However, the method proposed in this paper overcomes these limitations thanks to the thresholding used for binarization of the images. Indeed, in a backlit vision system the object will appear as a dark shadow over a bright field and the geometry being measured is the edge between the two areas. This edge is identified by a threshold which is not affected by intensity fluctuations if the contrast is high enough and the edge sharp enough as discussed above.

### 2.4. Measurement Campaign and Uncertainty Analysis

#### 2.4.1. Sample Preparation

To test the metrological performance of the designed backlit vision system we manufactured a set of samples of different average surface roughness (*R_a_*) through the machining of a C45 steel rod on a manual lathe. The rod was cut in pieces of same length and diameter of 12 mm, then the samples were turned on the same machine varying one of the process parameters and keeping the others constant.

In Figure 10 some of the variables involved in the turning process are shown:*V_t_* = cutting speed of the workpiece (m/min);*p* = depth of cut (mm);*f* = feed rate (mm/rev).

The variables that were kept constant were the cutting speed at 440 m/min and the depth of cut at 1 mm, while the feed rate was varied from 0.05 mm/rev to 0.6 mm/rev. This method enabled us to obtain a wide range of samples with different surface roughness, from 2.4 µm to 15.1 µm. In technical drawings of mechanical parts, *R_a_* values are usually specified as upper limits; less often they are indicated as an interval between upper and lower limit. Typical *R_a_* roughness achieved in turning processes is between 1.6 and 12.5 µm, therefore our experiment covers most of this range including *R_a_* = 3.2 µm (standard commercial machine finish), which is default roughness unless otherwise specified. The range of values used in this paper was selected based on literature regarding surface roughness measurements [21,27,28,29,40] and also because this range is of interest in common turning processes. Moreover, it covers a range wide enough to assess the performance of the instruments.

Table 1 shows the minimum and maximum roughness values measured with a stylus-based instrument with a tip radius of 2 µm and a tip angle of 60°; this was considered the reference instrument. In the last column, the range was computed as the maximum value minus the minimum value. The average values of *R_a_* were calculated on five different measurements obtained from different locations on each sample.

In Figure 11, pictures of three of the analyzed samples are shown, which are representative of the surface roughness range: sample 5, sample 2 and sample 7, respectively at *R_a_* 2.4 µm, 6.2 µm and 15.1 µm. The typical helicoidal grooves generated by the cutting tool on the rotating surface of the cylinder can be clearly observed. These grooves are parallel to each other, providing a rough surface with a quasi-periodic morphology. Observing the surface from a side, along a tangent direction, the series of valleys and peaks shown in Figure 3 will clearly appear to the observer, provided it has a sufficient spatial resolution.

#### 2.4.2. Comparison Analysis

The comparison analysis was conducted with the use of both contact and non-contact state-of-the-art measurement systems as follows (more information about the instruments can be found at the manufacturer’s website):a Surface Roughness Tester (Mitutoyo, S-3000), with a 2µm, 60° angle tip, referred to as SRT 1;a Surface Roughness Tester (Mitutoyo, SJ-210), with a 2µm, 60° angle tip, referred to as SRT 2;a confocal laser scanning microscope (Keyence, VK-X1000).

Each sample was measured with the three instruments and the results obtained were compared to the backlit vision instrument designed in this work. The measurements, even if different in nature, were performed in a way to obtain similar results:For the backlit vision system, to obtain one average surface roughness value, each sample was rotated 10 times and measured on different edges and the single value was calculated as the resulting average. This process was repeated five times;For the surface roughness testers, the samples were measured and subsequently rotated five times in order to collect measurements of different edges. The evaluation length (*ℓn*) on which the measurements were based was 15 mm;For the confocal microscope, the images collected were post-processed with its software (MultiFileAnalyzer) and the measurements were performed with a setting that would mimic the stylus-based instruments, on a *ℓn* = 15 mm as well.

At first, to simulate a procedure typical of real work environments, the measurements were performed on five different locations, not predetermined but evenly distributed on the surface. Later, we will see how these measurement conditions will influence the uncertainty associated with the measurement.

Details about the instruments’ resolution are reported in Table 2. It is important to mention that the resolution value relative to the backlit vision instrument is the pixel resolution, which is the corresponding pixel size in microns.

#### 2.4.3. Uncertainty Analysis Information

The performance of the backlit vision measurement system proposed in this study was compared to the other measurement instruments which are both contact and non-contact based. We based the performance comparison on the results obtained with the stylus-based instrument SRT 1, hence this was the reference instrument.

The ISO GUIDE 98-3 [41] is the main reference for the estimate of uncertainty, even if for industrial applications its complexity is being debated and simpler approaches are being proposed [42]. We carried out a Type A analysis of uncertainty and according to the ISO GUIDE 98-3 [41], given the limited number of samples available, we used the range as an estimate of the scatter of data instead of the standard deviation. The range is the difference between the highest and the lowest result.

## 3. Results

The results of the comparison analysis are presented in Figure 12. The samples are plotted in order of increasing roughness and the error bars represent two times the ranges associated with the measurements.

## 4. Discussion

By looking at the overview of the results in Figure 12, two things can be noticed:Two different behaviors can be seen based on the surface roughness: Samples 2 through 5 have a low surface roughness (between 2 µm and 6 µm), while Samples 1, 7, 8 and 9 have a higher surface roughness of around 14.5 µm;Regarding the uncertainty associated with each measurement (the uncertainty range), depicted in Figure 12, it can be said that in general the uncertainty is larger when measuring larger values of *R_a_*.

If we consider data reported in Figure 12 and limit the comparison to the backlit vision system and SRT 2, we can observe that results are compatible, i.e., the error bars are partially superimposed.

### 4.1. Critical Problems Related to the Uncertainty

When analyzing the data we pinpointed two reasons for the difference in uncertainty of measurement which we are going to discuss in this section:The location of the measurement on the surface of the turned part;The evaluation length on which the roughness measurement is based.

#### 4.1.1. Location of Measurement

Surface roughness is always calculated as an average value over a relatively short length (see Equation (1)), making the value intrinsically variable over the surface. In particular in turned pieces, the machining process itself creates an instability and a variability of the surface finish. Hence, if the measurements are not performed in the same location with all the instruments taking part in the experiment, the resulting values will have a higher variability due to both the variability of the surface itself and to the variability between the instruments. Using these values to compare the performance of different instruments may be questionable. Indeed, we miss a true reference, capable of providing a known input common to all instruments.

To address this hypothesis, we performed a test: we used the backlit vision instrument to again measure one sample, but in this case the measurements were performed always in the same location. The results of the test are reported in Figure 13.

Performing measurements in the same location allowed us to demonstrate that when performing measurements in the same location there is a noticeable decrease in the range associated with the results. The mean values are compatible, while the range associated with the random locations is significantly larger (1.0 µm) than the one associated with the same location (0.4 µm), which signifies that the variability in the part contributes significantly to disperse repeated measurements.

#### 4.1.2. Evaluation Length

The second reason for a difference in measurement uncertainty is the length on which the mean is evaluated (*ℓn*, evaluation length). In fact, it should increase with the increase of surface roughness.

For the contact instruments and for the confocal microscope, the evaluation length considered was *ℓn* = 15 mm. On the other hand, since the FoV of the designed backlit vision instrument is about 1.5 mm by 1.5 mm, this could be one of the reasons for its higher associated uncertainty. Each single measurement is calculated on the average of ten values measured on ten 1.5 mm by 1.5 mm images, which would equal the evaluation length considered for the other instruments. This has proven to be sufficient to obtain values which are in line with the other instruments, but it could also be one of the reasons its associated uncertainty is higher, especially on samples with higher *R_a_*.

To confirm the influence of the evaluation length, a second test was performed: a few samples were measured again with the confocal microscope but this time the surface roughness was based on five segments of *ℓn* = 1.5 mm. Then, these measurements were compared with the ones obtained on *ℓn* = 15 mm.

The results of the test are shown in Figure 14, and they highlight how the change in evaluation length has a clear impact on the range of the measurements. Dispersion decreases if evaluation length increases, as expected for the measurement of statistical quantities, such as *R_a_*.

### 4.2. Critical Problems Related to Higher R_a_s

The difference in behavior that can be noticed between lower and higher *R_a_*s could be caused by the difference in measuring techniques: the working principles of the instruments are different as might be the processing algorithms.

The most probable hypothesis concerns the size of the diamond tip of the stylus, which represents a limit to the instrument’s resolution. In fact, generally the error increases with the increase of the tip size or the increasing of the peak-to-valley height [43].

To discuss this hypothesis we developed a simple 2D geometrical model in which we simulate a roughness measurement performed with a stylus having a tip with radius 2 µm and a tip angle of 60°; in our experiments, both surface roughness testers had a tip with these characteristics. The physical dimension of the stylus tip prevents the probe from perfectly following the shape of the surface, especially when there is a sharp peak in roughness. The state of art in industrial application of styluses shows that a 2 µm tip is the lower limit to the probe tip radius [44]; in fact even recent literature studies the effect of stylus tip radius in *R_a_*. Our simple model is descriptive of best case scenarios in terms of tip radius in industrial applications.

In Figure 15 there are three step functions that represent the theoretical profile of three surfaces. We choose a step profile, because step response functions are generally useful to determine the performance of a measurement system. The model was meant to represent the concept of a sharp peak even if in real machined surfaces steps do not exist; the findings derived from it are descriptive of the situation that we observed at highest *R_a_s.* In particular, we simulate three steps having three increasing values of roughness: a 7 µm step, a 14 µm step and a 28 µm step. The corresponding *R_a_* theoretical values are exactly half of the step values, since the average surface roughness is measured as the distance of each point from a mean line (see Equation (1)), and they are reported in Table 3, column one.

The lines presented in the figure represent the simulation of the trajectory a stylus tip of the given dimensions (tip radius and angle) would follow when measuring a step profile: the shape of the tip prevents the stylus from reaching the bottom of the step transition, hence, a much softer trajectory is followed instead [7]. This is schematized in Figure 16. We traced the trajectory and used the corresponding values to calculate the surface roughness that the stylus would measure. The results are reported in Table 3, column two. When the step gets larger, the stylus underestimates *R_a_*. In particular, if we look at Δ*R_a_* we can notice how the difference increases with the increase of the surface roughness. Hence, this simple geometrical model shows how the stylus method tends to underestimate the real roughness value, especially in presence of sharp peaks which occur in case of large values of roughness.

This is a possible explanation of why the vision-based system that we have designed shows a different behavior, with respect to the reference instrument SRT1, when measuring higher *R_a_s* as compared to the lower *R_a_s*. While in the lower ranges of *R_a_* the two instruments are definitely consistent and in alignment, when moving to the large values of *R_a_*, the offset between the two increases and the stylus underestimates the roughness, which instead is correctly measured by the vision system. In fact, the backlit vision instrument is not limited by any kind of physical resolution if not only by the resolution of the optical system. Being a contactless sensor, no limitation arises from the physical contact of the probe to the target surface.

### 4.3. Second Set of Measurements

In Figure 17 the results of a new set of measurements are reported. In this case they were performed in the same location with the same evaluation length. It can be noticed that the uncertainty associated with the measurements performed by all of the instruments is much lower and that the error bars are partially superimposed.

The results show a general scatter of the data across all the different instruments and between the lower and higher *R_a_s*, making it difficult to state with certainty which instrument performs better compared to the reference one. Results show both bias and random scatter for all four instruments, so that none of them can be considered as a reference instrument from a metrological point of view.

Regarding the backlit vision system, the measurements of the samples with lower surface roughness are compatible with the results obtained with the surface roughness testers. While the measurements of the samples with higher surface roughness had a higher offset with respect to the surface roughness testers. The reason for this is the resolution limit of the stylus as explained in the model in Section 4.2.

## 5. Conclusions

In this paper we presented a backlit vision-based non-contact system to measure the surface roughness of a range of samples in a contactless mode with the purpose of in-line measurements.

The paper presents the design of the system, with attention towards:High magnification, fit to resolve *R_a_* in the range of 2–15 µm;An electronic control of focus through a tunable lens;White light for collimated backlighting.

After being assembled the system has been tested on typical cylindrical samples produced by turning a C45 steel rod on a manual lathe to obtain samples with *R_a_* in the range of interest.

The optimal imaging conditions were found by combining the use of the MTF and an ETL, which allows sharpening of the focus by controlling the current of the lens without repositioning the sample.

The measurements were based on the images of the samples acquired, then the average surface roughness *R_a_* was calculated thanks to the edge detection algorithm developed.

To evaluate the measurement uncertainty of the developed instrument, its performance was compared to the ones of other state-of-art roughness measurement systems. The measurement uncertainty was assessed by calculating the mean and uncertainty range associated with the measurement results, and the performance of each instrument has been compared to the chosen reference instrument, i.e., the stylus-based instrument SRT 1.

The conclusions derived from this comparison are as follows:The comparison of the results of the backlit system depends on the values of surface roughness considered;The measurements performed by the backlit vision system have a larger bias compared to the ones obtained by the stylus when measuring larger values of roughness, also because the stylus underestimates the *R_a_*;The results are compatible with the ones of the stylus at lower values of roughness. In fact, the error bands are superimposed by at least 58% based on the cases analyzed. The value was computed as percentage of overlap between the two uncertainty ranges with reference to the smaller one.

In conclusion:
The proposed instrument gives results which are comparable to the other state-of-art instruments when measuring lower surface roughness (2.4–6.2 µm), which are within the range normally achieved in turning, where the standard commercial machine finish is *R_a_* = 3.2 µm. This is important because it provides an innovative non-contact instrument for a potential application for ZDM quality control in many industrial turning processes;At higher values of surface roughness (14.3–15.1 µm) the offset with the reference instrument increases. Such high values are less frequent and less relevant for standard turning processes, however we tried to provide an explanation for this problem.

To provide an explanation of the problems encountered at the higher *R_a_* values, a simple geometrical model was developed, simulating a stylus measuring roughness through a surface having a step profile. The model confirms that large amplitude of the step determines the stylus to underestimate its *R_a_*. This observation shows a potential advantage of the backlit vision system: being non-contact, the measurement does not suffer any limitation due to the shape of the probe, while the stylus does.

Further studies will involve a more in-depth analysis of the influence of the location of the measurement and a study of the importance of the processing algorithms implemented by the different instruments.

We like to highlight that the developed backlit vision system for roughness measurement offers a setup that performs fast non-contact measurements, and in perspective, it opens the possibility of being implemented in the production line, allowing the inspection of 100% of production as required in ZDM.

## Figures and Tables

**Figure 1 sensors-23-01584-f001:**
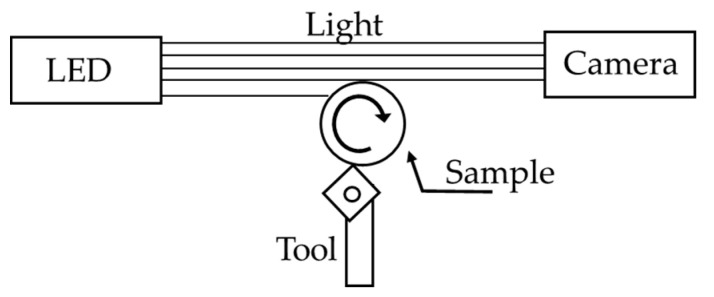
Simplified scheme of the conceived backlit vision-based measurement instrument where a cylindric bar is turned between a camera and a light source and surface roughness is measured in-line during production.

**Figure 2 sensors-23-01584-f002:**
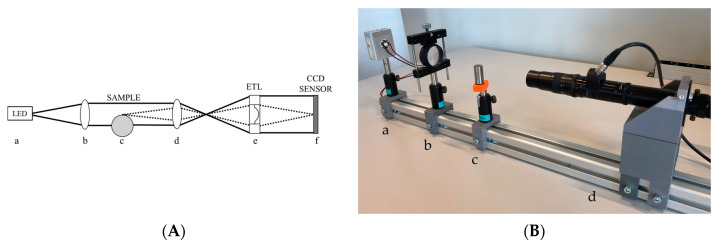
Measurement system setup. (**A**) is a schematic of the instrument composed as follows: (a) LED light, (b) lens, (c) test sample, (d) dynamic focus lens, (e) electrically tunable lens (ETL), (f) “2/3” CCD camera sensor. The black lines represent the light rays from the LED to the sensor, while the dashed lines are from an image point on the focus plane to the sensor. (**B**) is an image of the actual measurement setup: (a) LED light, (b) lens, (c) test sample, (d) objective.

**Figure 3 sensors-23-01584-f003:**
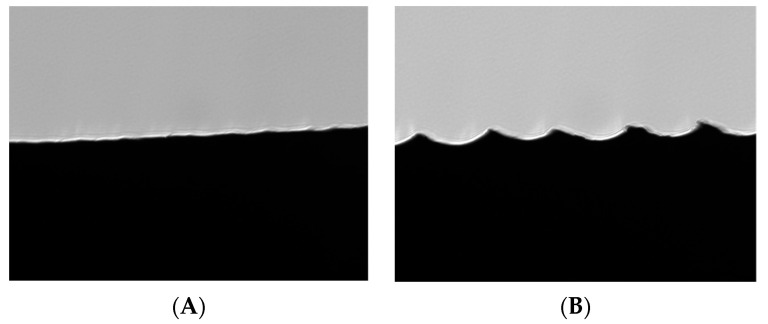
Acquired images of the edge of sample 5 (**A**) of *R_a_* = 2.4 µm and sample 7 (**B**) of *R_a_* = 15.1 µm. Full image size is 2448 × 2048 pixels corresponding to 1.4 × 1.7 mm.

**Figure 4 sensors-23-01584-f004:**
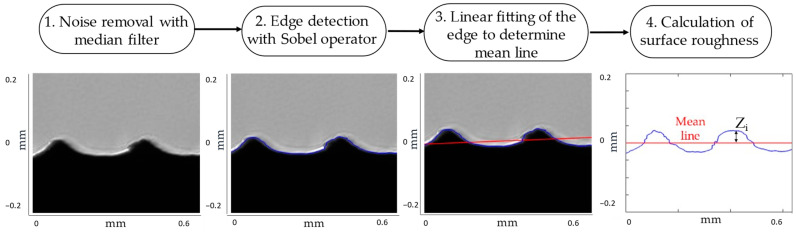
Steps of the measurement algorithm: (1) Filtering of the image; (2) The blue profile line is determined through the Sobel edge detection method; (3) The red line is the mean line obtained with the linear fit of the edge; (4) *R_a_* is measured as the average of the distances Z_i_ between each *i*th point of the profile and the mean line.

**Figure 5 sensors-23-01584-f005:**
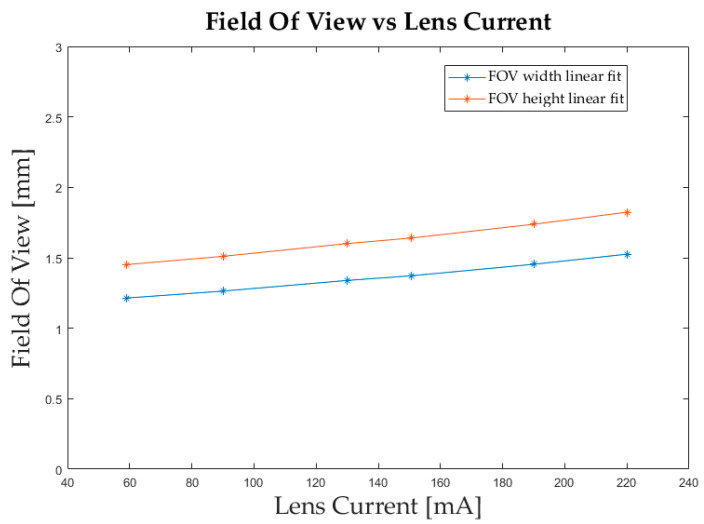
Linear correlation between the field of view (FoV) and the ETL current obtained through six calibration images.

**Figure 6 sensors-23-01584-f006:**
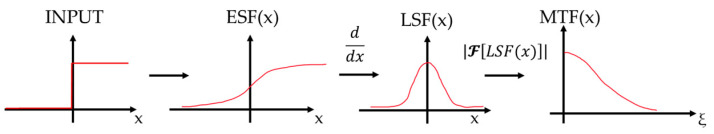
The response of the system to the step input is the edge spread function (ESF), then the spatial derivative of ESF data produces a line spread function (LSF), which is then Fourier transformed into the modulation transfer function (MTF).

**Figure 7 sensors-23-01584-f007:**
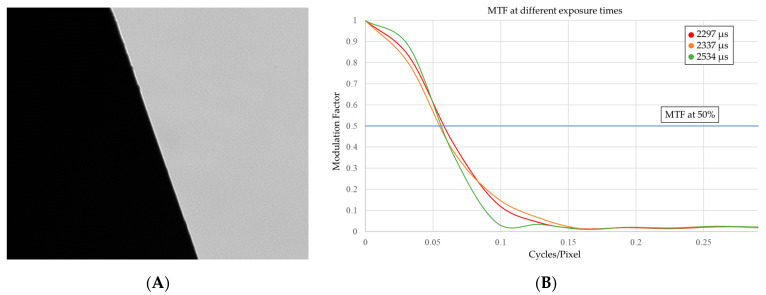
(**A**) Image of the sharp edge used as input; (**B**) MTF of the imaging system at different exposure times (the blue line is the MTF at 50%).

**Figure 8 sensors-23-01584-f008:**
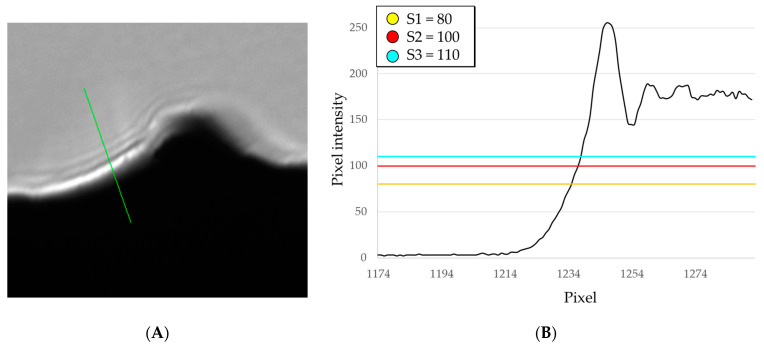
(**A**) The green line is an intensity profile orthogonal to the surface that intercepts the diffraction fringes. Image size is 350 × 350 pixels, corresponding to 0.2 × 0.2 mm; (**B**) Pixel intensity correlated to the green line: we see a transition from black to the first peak of the diffraction fringe followed by other smaller oscillations. Threshold values represent the cut-off values between black and white pixels in a binary representation of the image intensities: when the value changes the intensity transition is shifted either towards the black or towards the white.

**Figure 9 sensors-23-01584-f009:**
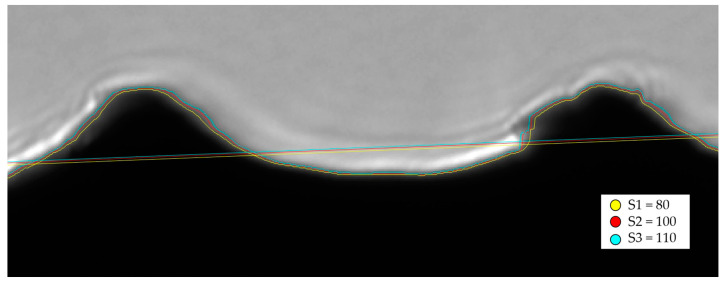
Comparison of different thresholds on the same close-up of the edge: the shape of the profile is maintained and only shifted, therefore *R_a_* is not affected. The three thresholds are described in the legend. Image size is 700 × 380 pixels, corresponding to 0.5 × 0.3 mm.

**Figure 10 sensors-23-01584-f010:**
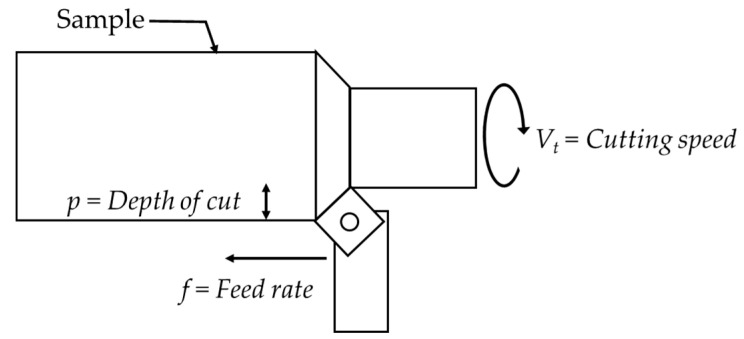
Basic turning parameters.

**Figure 11 sensors-23-01584-f011:**
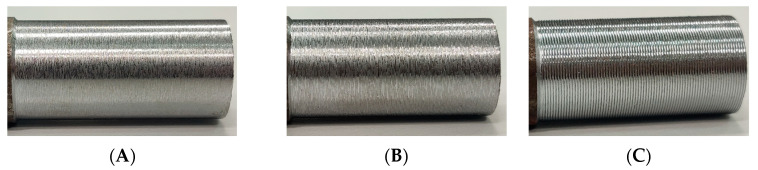
Surface roughness of sample 5 (**A**), sample 2 (**B**) and sample 7 (**C**).

**Figure 12 sensors-23-01584-f012:**
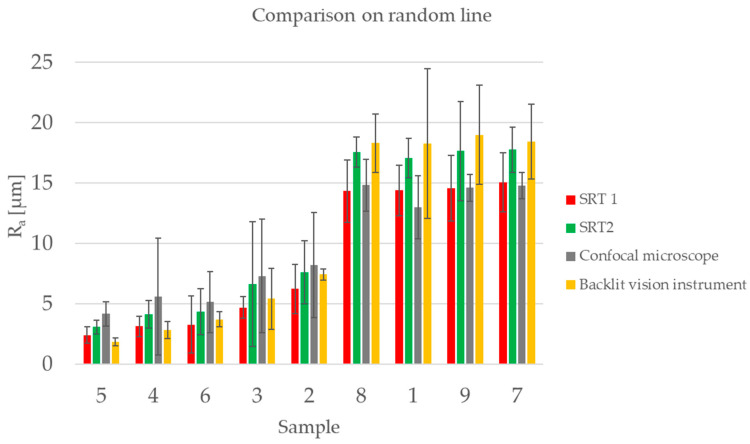
Overview of the results obtained by comparing the instruments on measurements performed on a random line on the surface. The samples are plotted in an increasing order of surface roughness of the reference instruments SRT 1. The error bar represents two times the range of the repeated measurements for each sample and each instrument.

**Figure 13 sensors-23-01584-f013:**
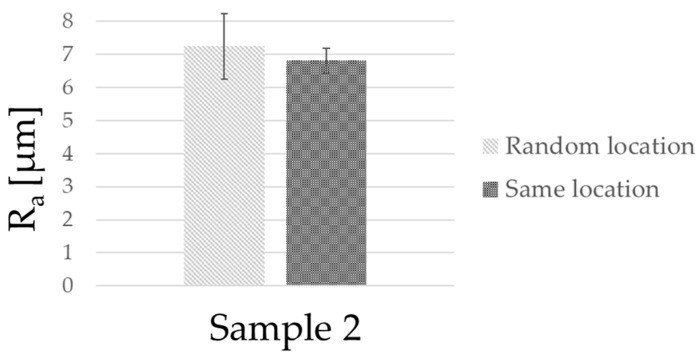
Comparison of results obtained by measuring surface roughness on different edges of the same sample with the VB instrument. The histogram on the left represents measurements performed in random locations, while the histogram on the right represents measurements taken on the same location. The error bars represent the range of the measurements.

**Figure 14 sensors-23-01584-f014:**
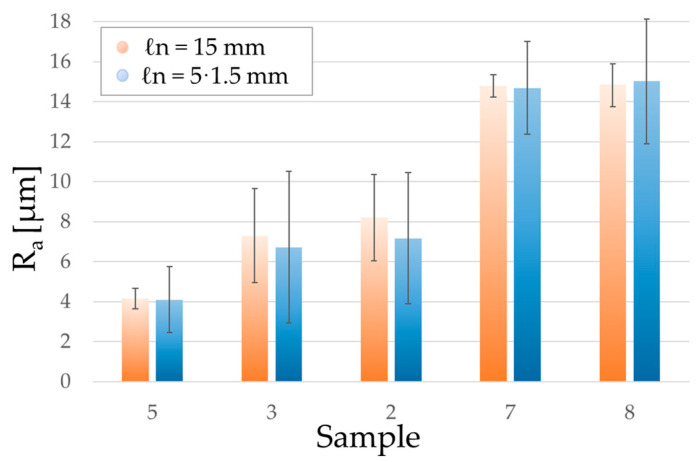
Comparison of results obtained by measuring surface roughness on different evaluation lengths (*ℓn*). The orange bars are the ranges associated with measurements taken on *ℓn* = 15 mm, while the blue bars are associated with five distinct measurements taken on *ℓn* = 1.5 mm.

**Figure 15 sensors-23-01584-f015:**
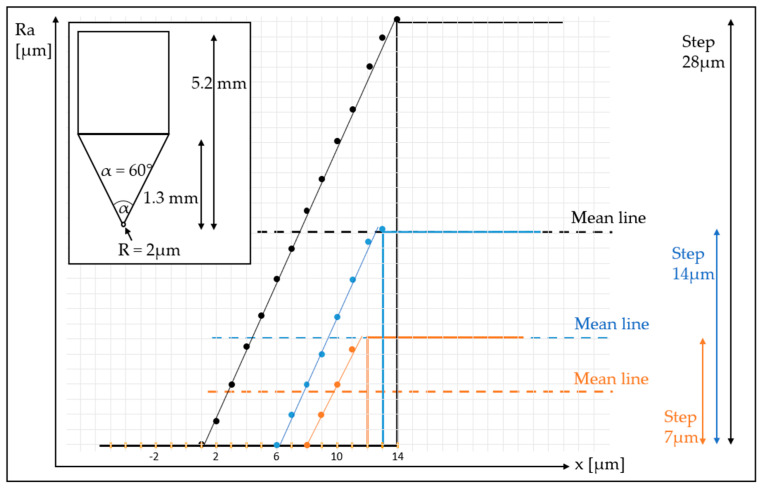
Simulation of a roughness measurement performed with a stylus’ three step functions of increasing amplitude. In the top left corner, there are the dimensions of the stylus tip: the tip radius is 2 µm and the tip angle is 60°.

**Figure 16 sensors-23-01584-f016:**
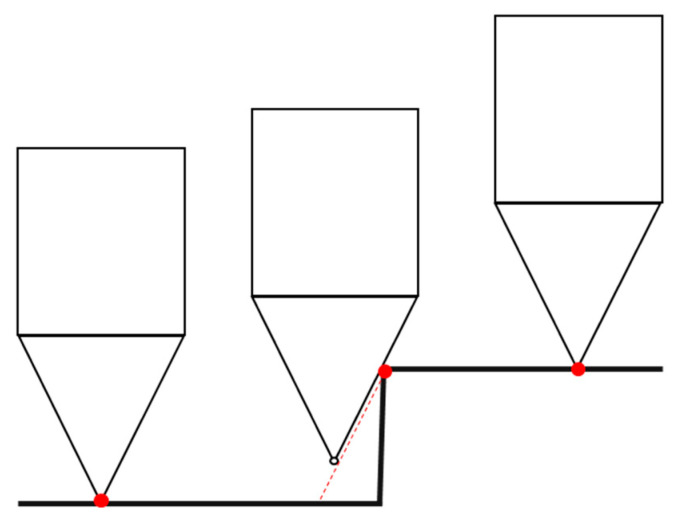
Effects of a finite stylus shape: the tip prevents the stylus from reaching the bottom of the step transition, the red trajectory is followed instead.

**Figure 17 sensors-23-01584-f017:**
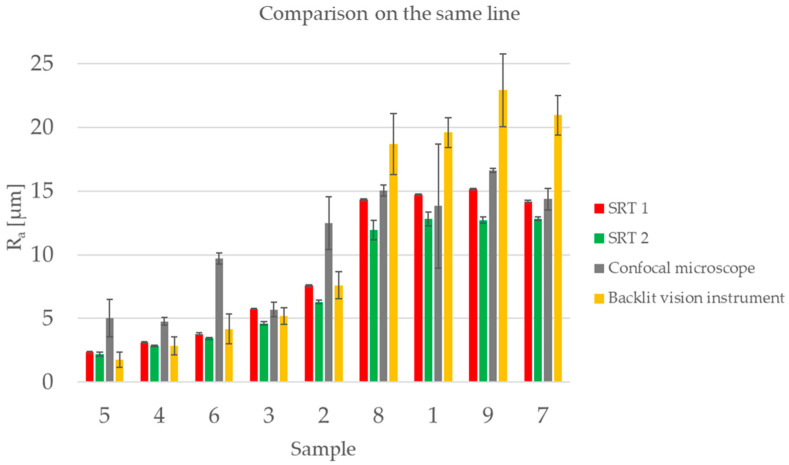
Overview of the results obtained by comparing the instruments on measurements performed on the same line on the surface. The samples are plotted in an increasing order of surface roughness of the reference instruments SRT 1. The error bar represents two times the range of the measurements.

**Table 1 sensors-23-01584-t001:** Surface roughness range measured with a stylus profilometer.

Sample	Avg *R_a_* (µm)	Min *R_a_* (µm)	Max *R_a_* (µm)	Range (µm)
1	14.4	13.7	14.7	1
2	6.2	5.7	6.7	1
3	4.7	4.4	4.9	0.5
4	3.1	2.9	3.3	0.4
5	2.4	2.3	2.6	0.3
6	3.3	2.6	3.8	1.2
7	15.1	14.3	15.5	1.2
8	14.3	13.9	15.2	1.3
9	14.6	13.9	15.3	1.4

**Table 2 sensors-23-01584-t002:** Summary of instruments.

Instr. ID	Type of Functioning	Resolution [µm]
1	Surface roughness tester 1 (SRT 1)	0.001 (for an 80 µm Z-range)
2	Surface roughness tester 2 (SRT 2)	0.002 (for a 25 µm Z-range)
3	Confocal microscope (CM)	±1.0 + L/100 (L = measuring length)
4	Backlit vision instrument (BV)	0.6–0.7 (µm/pixel) (pixel resolution range)

**Table 3 sensors-23-01584-t003:** Surface roughness of step function.

Reference Step (µm)	*R_a_* Measured by Stylus (µm)	Difference Δ*R_a_* (µm)
*R_a_* = 3.5	*R_a_* = 3.2	*R_a_* = −0.3
*R_a_* = 7	*R_a_* = 5.9	*R_a_* = −1.1
*R_a_* = 14	*R_a_* = 11.3	*R_a_* = −2.7

## Data Availability

Not applicable.

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
