# Peer review of "Design and Metrological Analysis of a Backlit Vision System for Surface Roughness Measurements of Turned Parts"

_sensors, 2023, doi:10.3390/s23031584_

Round 1

Reviewer 1 Report

In this paper,a backlit vision-based non-contact system to measure the surface roughness is designed and setup,  a set of turned steel samples with roughness in the range Ra 2.4 µm to 15.1 µm was measured. The feasibility of the proposed method is verified. The proposed method is innovative, but its practicality cannot fully support from the current data in the paper. Therefore, some key experiments, experimental results and discussions need to be further clarified. The specific problems are as follows:

1. The surface roughness of the sample range from Ra 2.4 µ m to 15.1 µ m. How was this value obtained? Meanwhile, what's the relationship between this roughness and ZDM?

2. In the abstract and conclusion, "the error bands are superimposed by at least 58%", How to get this value?

3 The literature review did not sufficiently introduce the state of the art of the research status and standards of surface roughness measurement methods in detail.

4. Figure 2, Figure 3, Figure 7, and Figure 8 are suggested to give scale bar.

5 Do not understand the "blurry fringes and the distinct fringes", in lines 245-246. What is the wavelength range of LED?  Meanwhile, add a filter to get monochromatic light is not so difficult, why not try. How does the uniformity of  the LED light source affect the image quality?

6 In Figure 8, the trend of the three lines is a little different from that of peak and valley.  It is recommended to give the roughness value based on the three lines respectively.

7. In section 2.4.2, it is recommended to provide the original pictures of the test and the original measurement data, otherwise it is difficult to judge the correctness of the experimental process and the results discussion.

8 In section 4.1, two reasons caused uncertainty, but no control method is proposed. If it can not be controlled, the practicability of visual measurement proposed in this paper is questionable. The deviation caused by position also exists in standard surface roughness measurement methods, but it is not so large. According to the examples, the uncertainty of the measurement results of this method is too large, which is an inherent problem of the method.

9 In Figure 14 and Figure 15, peak-valley difference is related to roughness , so the problem of 4.2 can be avoided if the stylus is suitable. The slit where the probe cannot fall down does not belong to the category of surface roughness.

10 In abstract, “short inspection time" is not mentioned in the paper.

Author Response

Please see attachment "Reviewer 1"

Reviewer 2 Report

Dear Author(s),

please find some comments on the manuscript ‘Design and Metrological Analysis of a Backlit Vision System for Surface Roughness Measurements of Turned Parts’, Manuscript ID: sensors-2148915:

1.      Generally, the ‘Introduction’ section is interesting, providing the required information. Nevertheless, the motivation and, respectively, critical review are missing. Especially, the motivation is not derived strictly from the review. Some limitations in other methods must be presented and then, correspondingly, some novelty proposed. Even though weaknesses in the measurement techniques are presented the motivation must be improved with a lack of knowledge. Straightly, the motivation is not received by the lack of knowledge of the current results.

2.      All of the set parameters presented in section 2.1. are not justified. Even obviously for a reader, their value selection justified that, currently, it looks like selected arbitrarily. The range of parameters must be justified, concluding.

3.      Could you please precise the ‘Sobel’ algorithm (lines 167-168) that, respectively, there are many Sobel proposals depending on the direction of the studies provided? If the algorithm fits the direction of sharp-edge detection, it should be maintained. It would be suitable if present the issue more precisely, especially with more details.

4.      Similarly to the previous comment, please try to present more consciously the mean (reference?) line with the linearly fitting method, lines 169-170. Moreover, it should be referenced if applied previously.

5.      According to the sentences ‘Among the possible methods to evaluate the MTF we chose the Slanted Edge Method: a step input is given to the imaging system through the image of a sharp slanted edge (see Figures 5 and 6 (a)), the system’s response is then the Edge Spread Function (ESF) which is then derived to obtain the Line Spread Function (LSF). Lastly the MTF is the Fourier Transform of the LSF [25]. To perform this analysis we used the Slanted Edge MTF Plugin of ImageJ open-source software package [25].’, lines 217-222, the usage of the ‘Slanted Edge MTF Plugin of ImageJ open-source software package’ software should be justified as well. Some advantages and, respectively, disadvantages of the usage of this software should also be addressed.

6.      Concerning the “thresholding’ method described in section 2.3.3., the accuracy of this method was not presented. How can be this concluded if the thresholding value is correct (suitable)? This was not studied if the thresholding value was modified. Please try to improve this technique with suitable clarification of the accuracy required.

7.      There are many variables presented in the body manuscript that, correspondingly, an additional section (e.g. Abbreviations/Shortcuts) seems to be required.

8.      There is suitable information provided for the measurement uncertainty, nevertheless, no words against the measurement noise were addressed. Please try to refer to that crucial issue, as follows:

(1)   https://www.doi.org/10.1117/1.OE.59.6.064110

(2)   https://www.doi.org/10.3390/s22030791

(3)   https://www.doi.org/10.1088/1361-6501/abb54f

9.      The ‘Conclusion’ section is interesting, nevertheless, it is difficult to find what are the most encouraging issues. This section should be divided into numbered gaps to highlight the most encouraging issue.

10.  Moreover, some additional suggestions to the ‘References’ section must be proposed:

-          Considering the references, there are many crucial items missed, especially from the ISO standards that treat measurement uncertainty.

-          I feel that some more up-to-date references should be placed that only 6 (from 28) are from the last 5 years. Especially the author(s) mentioned ‘recent literature’, line 89, but only the last ([19]) is recent, the rest ([15-18]) are not in my opinion. I would not define the paper published 15-20 years ago as ‘recent’, however, it is only my opinion.

-          Full reference data should be provided, e.g. ref. [9], there is no publisher or country. The same for authors, ‘et. al.’ should not be used but all of the authors mentioned.

Concluding, the manuscript reviewed seems to be interesting, and the area of study is up-to-date, definitely, nevertheless, some significant improvements must be provided before further processing of the manuscript reviewed.

Author Response

Please see attachment "Reviewer 2"

Round 2

Reviewer 1 Report

The author revised the paper according to the comments of reviewers, and the manuscript can be accepted.

One small suggestion, in line 237,"Matlab" should change to "MATLAB".

Reviewer 2 Report

Dear Authors,

according to the review of the revised manuscript titled Design and Metrological Analysis of a Backlit Vision System for Surface Roughness Measurements of Turned Parts’, Manuscript ID: sensors-2148915.

Thank you for improving so sophisticated and full responses to all of the raised comments.

It was found to improve suitably so, respectively, can be further considered for publication in the quality journal as the Sensors is.

All the best in future works.